# Quality of Life of Residents in Seniors’ Homes in Poland and Germany

**DOI:** 10.3390/healthcare12080829

**Published:** 2024-04-14

**Authors:** Katarzyna Swakowska, Dominik Olejniczak, Anna Staniszewska

**Affiliations:** 1Gute-Zeit24 Pflegedienst, 63785 Obernburg am Main, Germany; 2Department of Public Health, Medical University of Warsaw, 02-091 Warszawa, Poland; dominik.olejniczak@wum.edu.pl; 3Department of Experimental and Clinical Pharmacology, Medical University of Warsaw, 02-091 Warszawa, Poland; anna.staniszewska@wum.edu.pl

**Keywords:** quality of life, old age, disability, older adults, Poland, Germany

## Abstract

Background: Homes for the elderly and care facilities are not only a place of treatment, but also a place of permanent residence for older people. It is assumed that older adults’ quality of life in the centres may not be sufficient for their long well-being. The purpose of this study was to determine the level of quality of life among nursing home residents in Poland and Germany and the impact of disability on functioning in their major life domains. Material and Methods: This study was carried out using the WHOQOL-BREF questionnaire (abridged version) on 1000 people—500 residents of the centre in Poland and 500 residents in Germany. Results: The quality of life of Polish and German residents is at an average level and is closely related to their degree of independence. People with greater independence assessed their quality of life higher. Conclusion: The degree of disability affects one’s own health and the quality of life of the residents. Therefore, to improve older adults’ quality of life, certain steps should be taken, including supporting them in maintaining their health and independence on a daily basis.

## 1. Introduction

Until a dozen years ago, the issue of old age was raised much less frequently than it is today. The change has been driven primarily by an increase in life expectancy, accompanied by a decrease in the number of new births—all of which has led to changes in the demographic structure, the primary expression of which is a significant increase in the number of elderly people [1]. This, in turn, unequivocally affects the lifestyles of entire societies, as well as the problem of developing preventive health care and advances in the sphere of medical science. Already today, the increase in the number of elderly people has contributed to changes in the structure of demand for care and medical services, which will increase significantly within the next few years [2].

One of the manifestations of the changes indicated above is the increase in the number of people residing in senior citizen homes, which have now become not only a place to stay and receive treatment, but are also not infrequently—especially among societies in developed countries—treated as another stage of life, where an individual can encounter a wide variety of positive things. For this reason, it was decided to conduct a survey on quality of life among residents of such centres.

The financial status of German older adults is generally better than that of Polish counterparts. Still, the costs of staying at the specialized facility is much higher in Germany than in Poland. For that reason, numerous German seniors requiring specialized care choose to stay in Polish assisted living facilities, since they offer the lower cost of living and better medical care, especially in Polish private seniors’ homes, where the quality of care is better.

Both in Poland and Germany, the number of older adults is steadily increasing, making it necessary to introduce various systemic solutions in this regard. One of them is the introduction of institutional care [3]. At the moment, according to the data of 30 March 2022 in Germany, there are about 11,700 senior homes, which offer assistance to 915,000 seniors [4]. According to data from the Central Statistical Office as of 31 December 2021, there were 2015 facilities of this type in Poland; they offer support to 127.3 thousand wards [5].

### Quality of Life of the Older Adults

Because of the human life extension indicated above, it is necessary to focus on the problem of quality of life for older adults. Since old age is characterized by the systematic progression of various diseases, it is necessary to link quality of life to the treatment process and the ailments they have. In the case of older people, the ailments they have significantly affect their functional fitness, which in turn determines their degree of independence [6]. The indicated independence depends on a number of factors, including, first and foremost, the organic and functional changes that occur as a result of past illnesses, and access to facilities and medical services in general. Functional capacity is defined as the ability to perform basic life activities independently, and in turn is measured by the degree of independence and self-reliance [7,8].

Still, another element affecting the quality of life of older adults is the incidence of mental process disorders and depression among this group of people. According to a number of studies, depression in the over-65 population affects about 15–20% of people who see a primary care physician—additionally, it is the fourth leading cause of disability and risk of early death worldwide [9]. The next element affecting the quality of older adults’ life is their economic status—the financial resources they have, often insufficient, may contribute to their inability to adequately treat themselves and carry out all the doctor’s recommendations, because they simply cannot afford them [10].

The quality of life experienced by members of the social group in question is also influenced by the education they have, the number of friends they have, and the social life they lead, as well as such elements as owning an apartment outright [11].

Within the scope of this study, older adults’ quality of life is, therefore, perceived as a holistic overview of various aspects of an older individual’s existence within the care facility that gives an answer to the question whether such person is feeling well treated and cared for, while performing everyday tasks and activities related to self-care and leading a satisfactory life in different areas, such as social life, the environment, health issues, physical capabilities, and other essential parts of life.

## 2. Materials and Methods

The survey was conducted among residents of State Nursing Homes in Poland and in Germany, from January to December 2023. Initially, 1261 patients agreed to participate in this study. The remaining 261 surveys were rejected due to missing information. The final analysis covered surveys completed by 1000 patients; this corresponded to a total response rate of 79.3%. The study group consisted of 1000 respondents—500 residents from Poland and 500 residents from Germany. With such a sample size and the number of seniors’ home residents in Poland and in Germany (the numbers indicated in the Introduction), the error margin was 4.0% (95% confidence level and proportion 0.50). The required sample size was 384 subjects from Germany and 383 from Poland. The eligibility criteria for this study were that the person was 60 years of age or older (according to the World Health Organization, these people are classified as elderly), literate in the Polish or German language, any time of residence at a senior home, lack of memory and cognitive problems, and willingness to. We excluded residents who were under the age of 60, illiterate, not willing to participate, and with memory or cognitive disorders—confirmed by seniors’ home staff. Participation in the survey was voluntary and anonymous. Informed consent has been obtained from all participants included in this study.

This was a cross-sectional study. In order to obtain data of a quantitative nature, a diagnostic survey method was used—the research technique was a questionnaire on basic information about the respondents (age, gender, marital status, contact with family and friends, and willingness to live in a nursing home). The residents were asked to complete the questionnaire on their own; in addition, each of them was informed of the full anonymity of the survey being conducted. In situations where the respondent was unable to fill out the questionnaire they received on their own, an employee of the nursing home assisted them. Interviewers had experience in work with seniors and knew the realities of jobs in seniors’ homes (mean time of job in this facility: 8 yrs). The average time to complete the survey was 58 min.

This study also used the WHOQOL-BREF questionnaire, which is designed to make a subjective assessment of the quality of life of healthy people and sick people for clinical and cognitive valuation. The questionnaire consisted of 26 questions aiming to assess quality in four areas of life—physical, psychological, social, and environmental. The raw scores for each domain were calculated according to the key; then, the results were transformed so that they could be presented on a scale from 0 to 100. The higher the score, the higher the respondent’s quality of life in a given area. The questionnaire was analysed in 4 dimensions; in addition, two questions were analysed separately: the first question on the individual overall perception of quality of life and the second question on the overall perception of one’s own health. The Polish version of this questionnaire was prepared very professionally by K. Jaracz and L. Wołowicka from the Medical University of Poznań, in close cooperation with M. Kalfoss from the University of Oslo. Validity and reliability of this questionnaire (with Cronbach’s alpha coefficient > 0.7) are satisfactory [12]. Internal consistency of the German version of the WHOQOL-BREF, as measured with Cronbach’s alphas, of all subscales ranged between 0.57 and 0.88 [13].

Yet another tool used during this study was the Barthel scale, which determines the psychophysical fitness of the subject in ten aspects. Activities that determine the ability to function independently are assessed—eating, using the toilet, controlling the bladder sphincter, controlling the anal sphincter, bathing the entire body, dressing and undressing, moving (from bed to chair and back again, sitting down), moving on flat surfaces, and climbing and descending stairs [14]. The scale specifies three rating ranges: 0–20 points, total dependence; 20–80 points, partial lack of independence; 80–100 points means independence. Assessment of the usefulness of the Barthel questionnaire in Polish health care conditions as a reliable tool (Cronbach’s α coefficient = 0.78 ÷ 0.89; coefficient test–retest correlation R = 0.93 ÷ 0.95) to measure the scope of independence in performing life activities in everyday life by older people has been confirmed in research [15]. The reliability of the German version of the Barthel scale was excellent (mean kappa: 0.93) [16].

### 2.1. Ethics

This study was approved by the Ethics Committee, Medical University of Warsaw (No. AKBE/159/21). The respondents were advised about the purpose of this study. Participation was voluntary and anonymous. The completion of the survey meant that the patients gave their consent to participate in this study.

### 2.2. Statistical Research Methods

Spearman’s rank correlation coefficient was used to determine whether there were statistically significant correlational relationships between the ordinal and quantitative variables and the WHOQOL-BREF questionnaire scores. To determine whether there is a statistically significant difference in mean scores between the two study groups, the Student’s test was used, or if the assumption of normality of distributions is not met, the Mann–Whitney U test was used. A chi square test of independence was used to determine statistically significant correlations between the Barthel scale and the WHOQOL-BREF questionnaire scores. The conformity of the analysed distributions to the normal distribution was checked using the Shapiro–Wilk test. The statistical analyses assumed a significance level of *p* = 0.05. Analyses were performed using SPSS 24 software.

## 3. Results

### 3.1. Analysis of the Study Group

A group of 1000 people provided the survey’s results for the analysis as its participants. In total, 500 of them resided in the centres located in Poland, while the other 500 were in Germany. The group was noticeably dominated by the female share of 68% with the male respondents comprising the remaining 32%. Yet, the gender distributions were slightly different. In Poland, women made up 72% of the respondents with 28% male participation. In Germany, the number of women was lower—64%, while the men comprised 28% of the whole group. The gender differences between Polish and German study groups cannot be regarded as statistically significant with the score of *p* = 0.391.

For the whole survey, the average age of the residents was calculated as 68.4. The Polish participants’ average age was 69.0, while in the German centres, it was slightly lower—67.8. For the final results of this study, however, the difference in the average age does not seem to have much importance, as it coincides with *p* = 0.715.

The level of education for most of the participants was a vocational one (66%). The primary education level was acknowledged less frequently, comprising 17% of the group. The secondary education level made up 13% of the study assemblage. There was one person with a university degree (1%). The vocational education level proved to be the dominating one in both national groups with 60% in Polish centres and 72% in German facilities. Therefore, the statistics for the differences between those two did not show any significance for this study (*p* = 0.541).

A certain statistical significance was found in terms of the marital status of the participants in Poland and Germany—*p* = 0.012. The non-union respondents made up the majority of the study assemblage (94%), although their percentage in Germany was strikingly higher (100%) than of those in Polish facilities where 12% of respondents were in a relationship of some kind.

Within the length of stay at the facility, a standard deviation was identified as 2.63 years. Nevertheless, the average time was 4.1 years, yet the difference of the average stay time was measured as significantly considerable with the score of *p* = 0.019. The average length of facility inhabitance in the Polish group was considerably lower and counted as 3.4 years, while in Germany, the average number increased up to 4.7 years. Table 1 presents the socio-demographic characteristics of the study group.

### 3.2. Analysis of the WHOQOL-BREF Questionnaire

The first question of the WHOQOL-BREF Q was related to the participants’ perception of their life’s quality. For expressing the scores, a scale ranging from 1 to 5 was used. The average life’s quality was marked as 3.0. The standard deviation in this case accounted for 0.62. The 3 was a median score in this question. As Table 1 indicates, most of the study group’s participants (62%) did not see their lives’ quality as either good or bad, but rather as an average one. For 19% of respondents, it felt good, and similarly another 19% saw it as bad.

The same scale (1–5) was used for measuring the participants’ health satisfaction. It was slightly lower in this case, counting on average as 2.9, while the standard deviation here achieved 0.71. Still, the median score was counted as 3. The predominant part of the group (56%) did not show either satisfaction or dissatisfaction in the area of their health. There was, however, a difference in the extremes because the participation of dissatisfied people was larger (24%) than those who admitted they are satisfied in the area of their health. The most extreme responses (either very satisfied or very dissatisfied) equally obtained 1% of the share. The results are included in Table 2.

The results indicate that the feelings of the study’s participants in relation to their lives’ quality and health satisfaction are similar in both countries. Additionally, a notable correlation was found between one’s good or very good quality of life perception and having a high level of satisfaction in the health area. Namely, the participants feeling good about the life quality feel similarly about their health. The similar correlation was achieved in those who perceive their lives’ quality as poor as their satisfaction with health is adequately lower than of those having assessed their life quality as good.

The rest of the questions provided information regarding the four main spheres of quality of life. Those can be identified as physical, psychological, social, and environmental domains. The scale from 0 to 100 was used for measuring the results, which required certain transformation of data. The highest rates were attributed to the environmental area, closely followed by the psychological, social, and physical domains.

The social sphere related to the quality of life obtained the average of 54.3 points. In this case, the standard deviation was measured as 11.04 points, while the median number of points was 53.1. The range of scores’ distribution can be counted, therefore, as starting from 37.5 and reaching up to 84.4 p. The next one in the line, based on the results, was the psychological area of life and here the average equates to 51.6 points with the median score being 50.0. The standard deviation was counted as 12.70, while the range of distribution covered the section between 25.0 at the lowest and 87,5 at the highest. Within the area of social factors of the life’s quality, the average was 51.5 points. In this area, the standard deviation was measured as 12.20 points. The median number was 50.0, while the range of results started at 16.7 points and reached up to 91.7. The physical domain’s average number was 51.0. The counted median for this one was 50.0, with the standard deviation of 13.20 points. The range of results’ distribution accounted for between 28.6 and 85.7 points, which is indicated in Table 3.

The comparison of Polish and German residents did not indicate any significant statistical difference in their quality of life perception. The score was *p* = 0.053, which means that the differences do not affect the overall results in any statistically important way. In the health satisfaction area, the difference accounted for *p* = 0.068, while for the physical quality of life, it was *p* = 0.247. The result for the psychological area of life quality was *p* = 0.427 and for the social quality of life, it accounted for *p* = 0.311. Therefore, it can be noted that only one difference between Polish and German participants of this study was found, which was indicated as statistically significant, namely the one in the environmental area as it was *p* < 0.001. As it can be seen in Table 4, this particular domain of life quality was assessed noticeably higher by the residents of the Polish facilities.

All areas of participants’ life quality proved to be correlated in a statistically significant way. Those relations were found to be positive. The high scores in one area were reflected by similarly high scores in another area, which is indicated by the index *p* < 0.001. In the Polish facility, those relationships were exactly the same as in general results of this study. It was somehow different though for the German group; between physical and social areas of life quality, there were no statistically notable differences identified, while in the other positive correlations, the indicator was significant and amounted to *p* < 0.05.

The age and the participants’ assessment of their life quality as a whole correlated positively as indicated by the index *p* < 0.001 for the total number of the residents. Similarly, such correlations could be found in the individual sphere in the numbers of *p* = 0.002 for the physical domain and *p* < 0.001 for the psychological one, while the social area reached *p* < 0.021. The environmental domain’s index of correlation was *p* < 0.001. In the area of health satisfaction’s assessment, the index was at the point of *p* < 0.001. All the examined correspondences proved to be positive. Moreover, it was rather typical for the older residents to rate their quality of life relatively higher in every aspect that was verified, as indicated by Table 5.

When applied only to the Polish participants, the important statistical correlations were present between age and the general rating of the life quality (*p* = 0.002). The specific correlations for each of the specific areas were found to also be positive, and were as follows: *p* < 0.001 for the physical domain, *p* < 0.001 for psychological, *p* = 0.006 for social, and *p* < 0.001 for the environmental one. Additionally, the ratings in each area were increasing with age, being the highest for the older participants, as indicated in Table 6.

In the case of residents from Germany, there was no statistically significant correlation between age and WHOQOL-BREF questionnaire scores (*p* ≥ 0.05) (Table 7).

### 3.3. Barthel Scale

The analysis of the Barthel scale, on the other hand, indicated that the subjects differed in their range of functioning in daily life. Thus, among those surveyed, there were 15% of those who were independent, 45% who required partial assistance, and 40% of those who were dependent. For respondents from Polish centres, 20% were independent, 44% required partial assistance, 36% were dependent. In the case of residents from German centres, 10% were independent, 46% required partial assistance, 44% were dependent. Most respondents required partial assistance in daily functioning. The statistical analysis showed no significant differences between the study groups (*p* = 0.352) (Table 8).

Considering all the subjects, statistically significant correlations were found between the Barthel scale score and the evaluation of quality of life in general (*p* < 0.001) and in its individual domains, physical (*p* < 0.001), psychological (*p* < 0.001), social (*p* < 0.001), and environmental (*p* < 0.001), and the evaluation of satisfaction with health (*p* < 0.001). The correlation was negative in all cases; respondents who were more independent rated their quality of life higher in each area and rated their satisfaction with health higher (Table 9).

When considering residents from Poland, statistically significant correlations were found between the Barthel scale score and the assessment of quality of life overall (*p* < 0.001) and in its individual domains, physical (*p* < 0.001), psychological (*p* < 0.001), social (*p* < 0.001), and environmental (*p* < 0.001), and health satisfaction scores (*p* < 0.001). The correlation was negative in all cases; respondents who were more independent rated their quality of life higher in each area and rated their satisfaction with health higher (Table 10).

Considering the total number of subjects, statistically significant correlations were found between the Barthel scale score and the assessment of quality of life in general (*p* < 0.001) and in its individual domains, physical (*p* < 0.001), psychological (*p* < 0.001), social (*p* = 0.047), and environmental (*p* < 0.001), and the assessment of satisfaction with health (*p* < 0.001). The correlation was negative in all cases; respondents who were more independent rated their quality of life higher in each area and rated their satisfaction with health higher (Table 11). Referring to the above, it should be pointed out unequivocally that the health status and perceived ailments of the surveyed seniors have a real impact on the level of their quality of life; a correlation between these spheres was shown, both in the Polish and German groups.

Among those in a relationship, half were independent (50%) and half required partial assistance (50%). Among single people, only 13% were independent, 45% required partial assistance, and 43% were dependent. The difference is statistically significant (*p* = 0.020) (Table 12).

## 4. Discussion

Analysing the data obtained, it can be observed that the majority of people in both centres are single people largely requiring assistance in performing basic activities. All subjects achieved in the WHOQOL Q an average level of quality of life measurements. With the use of the Barthel scale, the analysis of the survey indicated that the life quality is generally rated as a less positive one by people who are less independent. This is additionally coinciding with their feelings regarding their health. There seems to be a powerful relation between health satisfaction and life quality. The lower people asses the first one, their perception of their lives’ quality is adequately lowered, too. The overall well-being and satisfaction were also affected largely by the quality of participants’ social life. In this area, residents in German centres acknowledged better feelings in the area of their social relations and contacts, which may indicate that this aspect of life is especially essential for their perception of the quality of life. The Polish residents acquired the lowest score in that area, although it did not noticeably affect the overall result, meaning that they still have an average quality of life, despite the fact that their social life is largely limited, especially in comparison with the German residents. On the other hand, the physical quality of life was rated lower by Germans, which may agree with the notion that the Polish public nursing homes offer better care than similar German public nursing homes, which results in better physical condition among the Polish centres’ residents.

The quality of life of respondents from both Poland and Germany was closely related to their degree of independence. Those with greater independence rated their quality of life higher. Such stronger correlations were found in Polish centres. The results obtained from our study are similar to those obtained by other authors conducting research on quality of life using other scales.

A survey of 411 nursing home residents in the area of the Silesian province using the SF-36v2 questionnaire confirmed that perceived pain largely affects functional performance and well-being. Respondents rated their functional status lowest, while mental health was rated highest [17]. Another study by Burzynska et al. conducted in the Lodz province among 117 residents provided information on the assessment of quality of life among residents of nursing homes, who rated their quality of life as neither good nor bad (56%); those rating their quality as bad and very bad were 26. 8% of respondents [18].

Other findings were obtained by authors regarding the Bialogard Nursing Home using the WHOQOL-BREF scale, where the highest score was obtained in the assessment of quality of life in the environmental and physical spheres, and the lowest in the social and psychological spheres. Those with greater independence rated their quality of life higher [19].

The study conducted by Chang et al. (2020) discovered the strong correlation between quality of life and depression and care dependency. It supports the statement that sustaining person-centred care in aged care settings is challenging and should be of high concern, leading to the development and application of interventions meant to reduce depression and care dependency among older adults [20].

Pramesona and Taneepanichskul (2018) researched the quality of life among older adults living in nursing homes due to compulsion. The study indicated that the level of quality of life is significantly lower among those forced to stay at such facilities in comparison to the others. The majority of those living in the nursing homes due to compulsion indicated that they did not receive proper care and support, had no social support resources, and suffered several chronic diseases, which affected their quality of life to a large extent [21].

Somewhat related to the issue of quality of life was the study performed by Slettebo and Polit (2008), which revealed that residents living in nursing homes admit that they feel safer there than they would be feeling living on their own. That was due to the uncertainty and fear that something could happen to them and they would not be able to receive help when needed. Therefore, having caregivers in close proximity seems to be one of the benefits of being in the nursing home [22].

Hara et al. (2022) researched the quality of life among Japan residents of nursing homes and found out that it was definitely worse than that of the general older population. Moreover, it tended to be especially low among males and those suffering specific diseases being under 65 years old. It also gradually decreased with the increasing level of dependency (care required for everyday existence). The scholars indicated that the measurement of the quality of life of nursing homes’ residents is essential for taking steps to improve and manage it [23].

According to Kane (2001), the main flaw that long-term care policies and programs in the United States suffer from is that they are balanced toward a model of nursing home care that tends to be associated with a poor quality of life for their users, while the costs are constantly raising [24].

Another study, conducted in Canada by Kehyayan et al. (2015), indicated that most residents of nursing homes are rather positive about various aspects of their quality of life, although there are also numerous aspects that require special attention because they seem to lower that positive feeling and satisfaction. For example, they did not find food and mealtime experiences as adding to their quality of life, but quite the contrary. It should be underlined that for the older adults residing in long-term care, food and eating are important factors affecting their quality of life perception. Dissatisfaction with food proves to be a serious problem, resulting often in reduced food intake, which leads to malnutrition risk, mortality, morbidity, and depression [25].

In analysing the research, it should be emphasized that quality of life is a complex issue that is difficult to clearly define. One of the factors that determine the quality of life of residents is not only providing round-the-clock care for residents, but also ensuring that there is an appropriate, perceived level of acceptance, belonging, and affiliation. An important issue is also to analyse the situation of elderly people living in DPS in terms of crises and the need to live in a care facility.

The differences between the residents of the Polish and German centres suggest, despite that they are not large in their scale, that even though the factors influencing the quality of life in the public nursing homes may seem similar, in different countries, the focus should be put more on different areas. It is related not only to the specifics of the nation and culture, but most of all to the specifics of the care offered in the centres. It was mentioned that some German older adults prefer to live in Polish nursing homes (15 participants of this performed study in the Polish nursing homes were actually Germans) due to the better care and lower cost. It may be advisable to address that issue in Germany. On the other hand, the social area of life in the Polish centres seems to be less satisfactory than in German ones, which indicates that this is an issue requiring much attention in Poland.

## 5. Conclusions

In the survey conducted, quality of life among residents in both Polish and German centres was at an average level, which indicates that in both cases, there should be some provided improvements to make the residents’ quality of life, despite their financial status and the costs of a given facility, increase as much as possible.

The specific needs of older adults, determining their perception of quality of life, should be addressed equally to those more general ones.

Maintaining and supporting independence and enabling older people to maintain as much autonomy as possible in everyday functioning is crucial to improving their well-being and quality of life.

Regular contact with other residents and the opportunity to participate in a variety of activities can significantly improve seniors’ well-being and overall quality of life.

A greater focus on providing social services to residents can have a significant impact on their mental and emotional well-being.

A more individualized approach is also needed, focusing on improving the physical conditions and comfort of residents.

It is also important to provide appropriate medical care and psychological support for older people who may suffer from various types of physical and mental health conditions. Access to specialist health services and rehabilitation therapies can significantly improve the quality of life of nursing home residents. In the context of cultural and social differences, it is also necessary to take into account the individual preferences and needs of residents. Some may prefer more privacy and independence, while others may find satisfaction in participating in group activities and social events. It is also important to involve care staff in the process of providing quality care and support for older people.

This study embraced a representative group of residents in seniors’ homes in Poland and in Germany, which is a strength point. However, this study has some limitations. First, we excluded residents who have cognitive disorders—confirmed by seniors’ home staff—that may affect the reliability of this study, because assessment staff could have been wrong. Second, chronic diseases can significantly affect the quality of life of older people; in our study, we did not ask about it.

## Figures and Tables

**Table 1 healthcare-12-00829-t001:** Sociodemographic characteristics of the study population.

		Country			*p*
		Poland	Germany	All	
Gender	Female	72.0%	64.0%	68.0%	0.391 ^1^
	Male	28.0%	36.0%	32.0%	
Age	Mean (±SD)	69.0 (±15.64)	67.8 (±15.97)	68.4 (±15.74)	0.715 ^2^
	Me	70.0	69.0	70.0	
	Min.	38.0	30.0	30.0	
	Max.	98.0	99.0	99.0	
Education	No answer	2.0%	4.0%	3.0%	0.541 ^1^
	Primary	20.0%	14.0%	17.0%	
	Vocational	60.0%	72.0%	66.0%	
	Secondary	16.0%	10.0%	13.0%	
	Higher	2.0%	0.0%	1.0%	
In relationship	Yes	12.0%	0.0%	6.0%	0.012 ^1^
	No	88.0%	100.0%	94.0%	
Residence time in senior home	Mean (±SD)	3.4 (±2.21)	4.7 (±2.87)	4.1 (±2.63)	0.019 ^3^
	Me	3.0	4.0	3.0	
	Min.	1.0	1.0	1.0	
	Max.	10.0	12.0	12.0	

^1^—statistics calculated using the chi square test of independence. ^2^—statistics calculated using Student’s *t* test. ^3^—statistics calculated using the Mann–Whitney U test. SD—Standard deviation; Me—Median; Min.—Minimum value; Max.—Maximum value.

**Table 2 healthcare-12-00829-t002:** Measures of central tendency and dispersion: participants’ quality of life and satisfaction with health.

Measurements (Scale 1–5)	Areas	
	Quality of Life	Health Satisfaction
Mean (±SD)	3.0 (±0.62)	2.9 (±0.71)
Me	3.0	3.0
Mo	3.0	3.0
Min.	2.0	1.0
Max.	4.0	5.0

SD—standard deviation; Me—median; Mo—dominant; Min.—minimum value; Max.—maximum value.

**Table 3 healthcare-12-00829-t003:** Basic statistics: quality of life four basic domains.

Measurements (Scale 1–100)	Areas			
	Environmental	Psychological	Social	Physical
Mean (±SD)	54.3 (±11.04)	51.6 (±12.70)	51.5 (±12.10)	51.0 (±13.20)
Me	53.1	50.0	50.0	50.0
Mo	50.0	50.0	50.0	46.4
Min.	37.5	25.0	16.7	28.6
Max.	84.4	87.5	91.7	85.7

SD—standard deviation; Me—median; Mo—dominant; Min.—minimum value; Max.—maximum value.

**Table 4 healthcare-12-00829-t004:** WHOQOL-BREF scale by study group Area/Resort.

Measurements		Areas					
		Quality of Life	Health Satisfaction	Environment	Psychological	Social	Physical
Poland	M (±SD)	3.1 (±0.66)	3.1 (±0.75)	53.4 (±16.03)	53.3 (±15.14)	52.8 (±13.53)	58.5 (±12.85)
Germany	M (±SD)	2.9 (±0.56)	2.8 (±0.64)	48.7 (±9.18)	49.9 (±9.54)	50.2 (±10.45)	50.2 (±6.78)
*p*		0.053	0.068	0.247	0.427	0.311	<0.001

**Table 5 healthcare-12-00829-t005:** Correlations between age and WHOQOL-BREF, population.

Age	Areas					
	Quality of Life	Health Satisfaction	Environment	Psychological	Social	Physical
Correlation coefficient	0.34 **	0.37 **	0.30 **	0.48 **	0.23 *	0.39 **
Significance (bilateral)	<0.001	<0.001	0.002	<0.001	0.021	<0.001

** Significant correlation at 0.01 (both sides). * Significant correlation at 0.05 (both sides).

**Table 6 healthcare-12-00829-t006:** Correlations between age and WHOQOL-BREF, Poland.

	Age	
	Correlation Coefficient	Significance (Bilateral)
Quality of life	0.44 **	0.002
Satisfaction with health	0.50 **	<0.001
Physical	0.51 **	<0.001
Physical	0.64 **	<0.001
Social	0.38 **	0.006
Environmental	0.57 **	<0.001

** Significant correlation at 0.01 (both sides).

**Table 7 healthcare-12-00829-t007:** Correlations between age and WHOQOL-BREF, Germany.

	Age	
	Correlation Coefficient	Significance (Bilateral)
Quality of life	0.25	0.078
Satisfaction with health	0.23	0.115
Physical	0.02	0.903
Physical	0.27	0.058
Social	0.05	0.727
Environmental	0.26	0.064

**Table 8 healthcare-12-00829-t008:** Barthel scale, comparison of centres in Poland and Germany.

			Resort		Total
			Poland	Germany	
Barthel Scale	Independent persons	Number	100	50	150
		% of Resort	20.0%	10.0%	15.0%
	Persons in partial need of assistance	Number	220	230	450
		% of Resort	44.0%	46.0%	45.0%
	Non-independent persons	Number	180	220	400
		% of Resort	36.0%	44.0%	40.0%
Chi square independence test			χ^2^ = 2.09; *p* = 0.352		

**Table 9 healthcare-12-00829-t009:** Correlations between Barthel scale and WHOQOL-BREF scale, total subjects.

Barthel Scale	Areas					
	Quality of Life	Health Satisfaction	Environment	Psychological	Social	Physical
Correlation coefficient	−0.73 **	−0.73 **	−0.78 **	−0.80 **	−0.56 **	−0.73 **
Significance (bilateral)	<0.001	<0.001	<0.001	<0.001	<0.001	<0.001

** Significant correlation at 0.01 (both sides).

**Table 10 healthcare-12-00829-t010:** Correlations between Barthel scale and WHOQOL-BREF scale, Poland.

	Barthel Scale	
	Correlation Coefficient	Significance (Bilateral)
Quality of life	−0.79 **	<0.001
Satisfaction with health	−0.79 **	<0.001
Physical	−0.88 **	<0.001
Physical	−0.88 **	<0.001
Social	−0.74 **	<0.001
Environmental	−0.83 **	<0.001

** Significant correlation at 0.01 (both sides).

**Table 11 healthcare-12-00829-t011:** Correlations between Barthel scale and WHOQOL-BREF scale, Germany.

	Barthel Scale	
	Correlation Coefficient	Significance (Bilateral)
Quality of life	−0.63 **	<0.001
Satisfaction with health	−0.64 **	<0.001
Physical	−0.61 **	<0.001
Physical	−0.71 **	<0.001
Social	−0.28 *	0.047
Environmental	−0.65 **	<0.001

** Significant correlation at 0.01 (both sides). * Significant correlation at 0.05 (both sides).

**Table 12 healthcare-12-00829-t012:** Barthel scale, comparison of people in a relationship and single.

			Relationship		Total
			Yes	No	
Barthel Scale	Independent persons	Number	30	120	150
		% of Resort	50.0%	12.8%	15.0%
	Persons in partial need of assistance	Number	30	420	450
		% of Resort	50.0%	44.7%	45.0%
	Non-independent persons	Number	0	400	400
		% of Resort	0.0%	42.6%	40.0%
Chi square independence test			χ^2^ = 7.80; *p* = 0.020		

## Data Availability

Data are contained within the article.

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
