# Peer review of "Quality of Life of Residents in Seniors’ Homes in Poland and Germany"

_healthcare, 2024, doi:10.3390/healthcare12080829_

Round 1

Reviewer 1 Report

Comments and Suggestions for Authors

Dear authors,

The manuscript is interesting and my suggestions are to improve clarity and readability. Please, consider them all.

- The introduction is way too large. Be more concise on all topics.

- I could not notice the sample size calculation. This is essential for any further inferences.

- The participants' characteristics could be shown using a table. Also, some characteristics and comparisons are missing: average age, anthropometrical measures, marital status, and other variables that could be potential counfounders or co-varables for all your statistics.

- The methods are also large. Please, restrict to the essential about WHOQOL. Psychometric characteristics are missing.

- The discussion should be expanded. There are several major results that were not well discussed. The section must be improved and the references from other studies added.

Author Response

Dear Editor

Thank you very much for all the comments and suggestions provided by your Reviewers. It is of great importance to me that I can take advantage of the opinions of experienced colleagues who have devoted their time to carefully and thoroughly evaluate a manuscript submitted to Healthcare. I would also like to thank the Editorial Board for the opportunity to submit a corrected version of the manuscript entitled Quality of life of residents in seniors' homes in Poland and Germany. I hope that the additions and improvements made to the text will make the text more interesting for the journal's international audience.

We have responded to each of the Reviewers’ comments and have incorporated all modifications suggested by the Reviewers into the revised manuscript. The changes within the revised manuscript were made in change tracking mode. Our responses to the Reviewers’ comments are as follows:

Reviewer 1

Reviewer’s comment 1: The manuscript is interesting and my suggestions are to improve clarity and readability. Please, consider them all.

Our answer #R1-1: Thanks you for your opinion.

Reviewer’s comment 2:  The introduction is way too large. Be more concise on all topics.

Our answer #R1-2:  Removed unnecessary parts of the introduction.

Reviewer’s comment 3: I could not notice the sample size calculation. This is essential for any further inferences.

Our answer #R1-3:  Inserted.

Reviewer’s comment 4:  The participants' characteristics could be shown using a table. Also, some characteristics and comparisons are missing: average age, anthropometrical measures, marital status, and other variables that could be potential counfounders or co-varables for all your statistics.

Our answer #R1-4: Inserted Table 1 with socio-demographic characteristics.

Reviewer’s comment 5: The methods are also large. Please, restrict to the essential about WHOQOL. Psychometric characteristics are missing.

Our answer #R1-5: Some information about the tool WHOQOL has been removed. Inserted psychometric characteristics.

Reviewer’s comment 6:  The discussion should be expanded. There are several major results that were not well discussed. The section must be improved and the references from other studies added.

Our answer #R1-6: Following the suggestions of reviewer 2, the discussion was reorganized and some fragments of the introduction were moved to the discussion.

Reviewer 2 Report

Comments and Suggestions for Authors

Thank you very much for giving me the opportunity to review this work. Below I present some questions with the aim of helping to improve the presentation of the work carried out.

CONSIDERATIONS FOR AUTHORS

L 52-59. The contents reflected here are more related to results and strategies to be used based on those results than to giving a vision of the current situation of the topic to be discussed.

L154-200 the data referred to there would be appropriate to use within the discussion section.

No inclusion/exclusion criteria section included.

No type of design is specified.

Nor is it explained how the administration of the questionnaires was carried out, they were all administered at the same time, separately, with/without the help of a professional knowledgeable about them,…

Table 2 and Table 3. The standard deviation usually appears in parentheses next to the mean score.

Table 4, Table 5 and 6. How was age expressed to establish the correlation? It should be explained.

Why are the scores achieved by the participants not used to study the BI/WHOQOL correlations?

The conclusion must be more concrete and in accordance with the objectives. The data included could be in the discussion.

Author Response

Dear Editor

Thank you very much for all the comments and suggestions provided by your Reviewers. It is of great importance to me that I can take advantage of the opinions of experienced colleagues who have devoted their time to carefully and thoroughly evaluate a manuscript submitted to Healthcare. I would also like to thank the Editorial Board for the opportunity to submit a corrected version of the manuscript entitled Quality of life of residents in seniors' homes in Poland and Germany. I hope that the additions and improvements made to the text will make the text more interesting for the journal's international audience.

We have responded to each of the Reviewers’ comments and have incorporated all modifications suggested by the Reviewers into the revised manuscript. The changes within the revised manuscript were made in change tracking mode. Our responses to the Reviewers’ comments are as follows:

Reviewer 2

Reviewer’s comment 1: Thank you very much for giving me the opportunity to review this work. Below I present some questions with the aim of helping to improve the presentation of the work carried out.

Our answer #R2-1: Thanks you for your opinion.

Reviewer’s comment 2:  L 52-59. The contents reflected here are more related to results and strategies to be used based on those results than to giving a vision of the current situation of the topic to be discussed.

Our answer #R2-2: Fragment L52-59 has been moved to the conclusions.

Reviewer’s comment 3: L154-200 the data referred to there would be appropriate to use within the discussion section.

Our answer #R2-3: Fragment L154-200 has been moved to the conclusions.

Reviewer’s comment 4:  No inclusion/exclusion criteria section included.

Our answer #R2-4: Inserted.

Reviewer’s comment 5: No type of design is specified.

Our answer #R2-5: Inserted.

Reviewer’s comment 6:  Nor is it explained how the administration of the questionnaires was carried out, they were all administered at the same time, separately, with/without the help of a professional knowledgeable about them,…

Our answer #R2-6: In paper was written “The residents were asked to complete the  questionnaire on their own,  in addition,  each of them was informed of the full anonymity of the survey being conducted.  In situations where the respondent was  unable to fill out the questionnaire he received on his own,  an employee of the nursing home assisted him.” Inserted: Interviewers had experience in work whit seniors and knew the realities of job in seniors homes (mean time of job in this facility 8 yrs). The average time to complete the survey was 55 minutes.

Reviewer’s comment 7:  Table 2 and Table 3. The standard deviation usually appears in parentheses next to the mean score.

Our answer #R2-7: Changed in Table 1, Table 2 and Table 3.

Reviewer’s comment 8:  Table 4, Table 5 and 6. How was age expressed to establish the correlation? It should be explained.

Our answer #R2-8: Age was expressed on a quantitative scale, the first table gave the mean age with standard deviations. Both in Poland and in Germany it was consistent with the normal distribution.

Reviewer’s comment 9:  Why are the scores achieved by the participants not used to study the BI/WHOQOL correlations?

Our answer #R2-9: The WHOQL results did not meet the assumption of compliance with the normal distribution. Therefore, Spearman's rank correlations were used. The correlation was positive in all cases where it was significant, and the respondents in older age rated their quality of life higher in particular areas. Such results were for the general public and for Polish, in the case of Germany no such significant relationships were found.

Reviewer’s comment 10:  The conclusion must be more concrete and in accordance with the objectives. The data included could be in the discussion.

Our answer #R2-10: The analysis of the correlation between the degree of autonomy of an individual and his or her ability to self-care and the subjective assessment of the quality of life was extended. The debate on the provision of social care and adequate support and medical care to residents has been expanded.

General note: Is changed numbering references, because reorganized parts of the manuscript.

Round 2

Reviewer 1 Report

Comments and Suggestions for Authors

Dear authors,

Thank you for addressing all reported issues.

Regards.

Author Response

Thank you very much .

Reviewer 2 Report

Comments and Suggestions for Authors

Dear Authors,

I consider that you have worked on the article, but more changes need to be made to improve its presentation and acceptability by readers.

Our answer #R2-2: Fragment L52-59 has been moved to the conclusions. The conclusion paragraph should be concise.

Our answer #R2-3: Fragment L154-200 has been moved to the conclusions. The conclusion paragraph should be concise

Our answer #R2-4: Inserted. Memory and/or cognition disorders must be confirmed using some measuring instrument: scale, questionnaire,... it is not useful for the nursing home workers to confirm it.

Our answer #R2-7: Changed in Table 1, Table 2 and Table 3. .  Example: 69.00 (±15.64), on the same line and this for all tables

Age: 70.0 (30.0-98.0) all in the same line

Our answer #R2-10: The analysis of the correlation between the degree of autonomy of an individual and his or her ability to self-care and the subjective assessment of the quality of life was extended. The debate on the provision of social care and adequate support and medical care to residents has been expanded.

L552 to 571 are not conclusions, they are content to include in discussion.

L 573, may be a conclusion of the study.

You cannot present 90 lines of conclusions, the conclusions need to be drawn. 

Author Response

Dear Reviewer, thank you for your time and valuable comments. We tried to respond to them as best as possible. Below we present the answers to your tips.

Our answer #R2-2: Fragment L52-59 has been moved to the conclusions. The conclusion paragraph should be concise.

Answer 2: This fragment has been removed.

Our answer #R2-3: Fragment L154-200 has been moved to the conclusions. The conclusion paragraph should be concise

Answer 2: Small fragment has been removed, the rest is in our opinion important.

Our answer #R2-4: Inserted. Memory and/or cognition disorders must be confirmed using some measuring instrument: scale, questionnaire,... it is not useful for the nursing home workers to confirm it.

Answer 2: Dear reviewer, the study has ended, so we are unable to evaluate seniors in this respect. As we wrote in the text, these disorders were not detected in residents, which was confirmed by employees based on medical examination and daily observation of seniors.

Our answer #R2-7: Changed in Table 1, Table 2 and Table 3. .  Example: 69.00 (±15.64), on the same line and this for all tables

Age: 70.0 (30.0-98.0) all in the same line

Answer 2: Changed.

Our answer #R2-10: The analysis of the correlation between the degree of autonomy of an individual and his or her ability to self-care and the subjective assessment of the quality of life was extended. The debate on the provision of social care and adequate support and medical care to residents has been expanded.

L552 to 571 are not conclusions, they are content to include in discussion.

L 573, may be a conclusion of the study.

You cannot present 90 lines of conclusions, the conclusions need to be drawn. 

Answer 2: We decided to remove some of the text - this is in line with the reviewer's suggestion. We have presented more concise conclusions.

Round 3

Reviewer 2 Report

Comments and Suggestions for Authors

What does it mean?

TABLE 1.  Age. Mean(±SD). [.1]

                 Residence time in senior home [.2]

TABLE 2. [.3]

TABLE 3. [.4]

Line 354. [.5]

Line 400. [.6]

Author Response

Dear reviewer, I don't understand your question. Perhaps these are changes that are visible when tracking changes, after accepting the text, the changes you presented are not visible, please look at the attached text.
